# Suppression of the ABCA1 Cholesterol Transporter Impairs the Growth and Migration of Epithelial Ovarian Cancer

**DOI:** 10.3390/cancers14081878

**Published:** 2022-04-08

**Authors:** Jixuan Gao, MoonSun Jung, Rebekka T. Williams, Danica Hui, Amanda J. Russell, Andrea J. Naim, Alvin Kamili, Molly Clifton, Angelika Bongers, Chelsea Mayoh, Gwo Ho, Clare L. Scott, Wendy Jessup, Michelle Haber, Murray D. Norris, Michelle J. Henderson

**Affiliations:** 1Children’s Cancer Institute, Lowy Cancer Research Centre, UNSW Sydney, Sydney, NSW 2052, Australia; mjung@ccia.org.au (M.J.); rebekka.t.williams@gmail.com (R.T.W.); danicahui96@gmail.com (D.H.); a.russell@garvan.org.au (A.J.R.); andrea.j.naim@gmail.com (A.J.N.); akamili@ccia.org.au (A.K.); cliftonmolly@gmail.com (M.C.); abongers@ccia.org.au (A.B.); cmayoh@ccia.org.au (C.M.); mhaber@ccia.org.au (M.H.); mnorris@ccia.org.au (M.D.N.); mhenderson@ccia.org.au (M.J.H.); 2Telomere Length Regulation Unit, Children’s Medical Research Institute, Westmead, NSW 2145, Australia; 3School of Women’s and Children’s Health, UNSW Sydney, Sydney, NSW 2052, Australia; 4Garvan Institute of Medical Research, Sydney, NSW 2010, Australia; 5Australia Walter and Eliza Hall Institute of Medical Research, Melbourne, VIC 3052, Australia; ho.g@wehi.edu.au (G.H.); scottc@wehi.edu.au (C.L.S.); 6ANZAC Research Institute, Concord, Sydney, NSW 2139, Australia; wendy.jessup@gmail.com; 7UNSW Centre for Childhood Cancer Research, UNSW Sydney, Sydney, NSW 2052, Australia; 8Peter MacCallum Cancer Centre, Melbourne, VIC 3000, Australia; aocs@petermac.org

**Keywords:** epithelial ovarian cancer, ABCA1, cholesterol

## Abstract

**Simple Summary:**

Epithelial ovarian cancer (EOC) is the most lethal gynaecological cancer. Over 80% of cases have already spread at diagnosis, and these patients face a five-year survival rate of 35%. EOC cells often spread to the greater omentum, an abdominal fat pad. Here, EOC cells take-up cholesterols. Excessive amounts of cholesterol are lethal; thus, we proposed that the ABCA1 cholesterol transporter exports cholesterol from serous EOC cells to maintain cholesterol balance. Indeed, we found that reducing the level of ABCA1 could suppress serous EOC growth in two-dimensional as well as three-dimensional cell culture and also hindered their migration, a key process required for cancer spread. We also identified drugs that impair EOC cell growth by inhibiting cholesterol export. Our data demonstrate that disrupting the cholesterol balance by targeting ABCA1 may be an effective treatment strategy for EOC patients.

**Abstract:**

Background: Epithelial ovarian cancer (EOC) is the most lethal gynaecological malignancy with over 80% of cases already disseminated at diagnosis and facing a dismal five-year survival rate of 35%. EOC cells often spread to the greater omentum where they take-up cholesterol. Excessive amounts of cholesterol can be cytocidal, suggesting that cholesterol efflux through transporters may be important to maintain homeostasis, and this may explain the observation that high expression of the ATP-binding cassette A1 (ABCA1) cholesterol transporter has been associated with poor outcome in EOC patients. Methods: ABCA1 expression was silenced in EOC cells to investigate the effect of inhibiting cholesterol efflux on EOC biology through growth and migration assays, three-dimensional spheroid culture and cholesterol quantification. Results: ABCA1 suppression significantly reduced the growth, motility and colony formation of EOC cell lines as well as the size of EOC spheroids, whilst stimulating expression of ABCA1 reversed these effects. In serous EOC cells, ABCA1 suppression induced accumulation of cholesterol. Lowering cholesterol levels using methyl-B-cyclodextrin rescued the effect of ABCA1 suppression, restoring EOC growth. Furthermore, we identified FDA-approved agents that induced cholesterol accumulation and elicited cytocidal effects in EOC cells. Conclusions: Our data demonstrate the importance of ABCA1 in maintaining cholesterol balance and malignant properties in EOC cells, highlighting its potential as a therapeutic target for this disease.

## 1. Introduction

With a five-year survival rate for patients with advanced-stage disease of only 35%, epithelial ovarian cancer (EOC) is the most lethal gynaecological malignancy [1]. EOCs arise from the ovarian epithelium on the external surface of the ovaries and can be subdivided into five histological subtypes, namely, high-grade serous (HG-SOC; accounting for 70% of all cases), low-grade serous, endometrioid, clear-cell and mucinous tumours [2,3]. HG-SOCs are thought to arise from the ciliated cells of the fallopian tube, are highly proliferative and 80% of cases are TP53 mutant [3,4]. Low-grade serous EOCs are much rarer and usually do not exhibit TP53 mutations [3,4]. Endometrioid EOCs account for 10–20% of all EOCs and exhibit a range of somatic mutations including ARID1A and PTEN [3,4]. ARID1A mutations are also often seen in clear cell carcinoma, which accounts for ~5% of all EOCs [3,4]. Mucinous is the rarest subtype, accounting for 2–3% of EOCs and characterized by KRAS mutations [3,4]. High-grade serous and endometrioid ovarian cancers have the poorest clinical prognosis compared to other subtypes [3,4]. Treatment for EOC patients typically begins with tumour debulking surgery followed by a combination of carboplatin and paclitaxel chemotherapeutics [4,5,6]. Despite the use of intensive chemotherapy, the survival rates of EOC patients have barely improved over the past several decades, prompting the need for more effective, less toxic treatments.

One reason for treatment failure in EOC is that over 80% of EOC patients already have aggressive, disseminated (stage 3 or 4) disease at the time of diagnosis [7]. Trans-coelomic dissemination occurs through shedding of EOC cells into the peritoneal ascitic fluid, which acts as a medium for EOC migration to the abdominal organs, peritoneum and greater omentum [8]. The greater omentum is a common site of spread for EOC because omental adipocytes and adipose-derived stem cells (ADSCs) secrete adipokines and growth factors, which induce the homing of EOC cells and enable rapid tumour growth [9,10,11]. Here, ovarian cancer cells can uptake cholesterol and other lipids from their extracellular environment resulting in altered lipid metabolism [9,12,13,14]. In the context of cancer biology, cholesterol has generally been implicated in pro-malignant roles such as forming lipid rafts to enable heightened expression of growth-promoting receptors [9,15]. However, the accumulation of cellular cholesterol can be cytocidal, a phenomenon that has best been demonstrated in macrophages where excessive amounts of cholesterol has led to apoptosis [16,17,18,19,20]. This suggests that cells, such as EOC cells, that exist in a cholesterol-rich environment could have a critical need for cholesterol balance, which could be achieved in part through the action of membrane-bound cholesterol efflux transporters.

In triple-negative breast cancer, another gynaecological malignancy that thrives in a cholesterol-rich environment, impairing cholesterol efflux through suppression of the ATP-binding cassette A1 (ABCA1) cholesterol transporter leads to disruption of cholesterol homeostasis and impaired malignant characteristics [21,22], and high expression of ABCA1 is associated with high-grade tumours in this disease [14]. In EOC, we previously found elevated ABCA1 expression to be associated with poor clinical outcome [23], suggesting a role for cholesterol efflux in this disease as well.

We conducted a detailed investigation into the role of ABCA1 in cholesterol balance in EOC. For the first time, we determined the impact of ABCA1 suppression on intracellular cholesterol balance, cell growth and migration in EOC cells and on the characteristics of three-dimensional EOC spheroids.

## 2. Materials and Methods

### 2.1. Patient Cohorts

Samples from 150 patients with serous EOC and 61 patients with endometrioid EOC were obtained from the Australian Ovarian Cancer Study (AOCS), a population-based case-controlled study undertaken between 2002 and 2006. An additional 19 endometrioid EOC samples were obtained from the Gynaecological Oncology Biobank at Westmead Hospital (GynBiobank), Sydney, Australia. Fresh frozen tumour specimens were obtained at the time of surgical debulking, prior to chemotherapeutic exposure. Samples were verified as containing at least 70% tumour tissue. Both the serous and endometrioid EOC cohorts consisted of tumours from all surgical stages (according to FIGO classification) with a median clinical follow up of 46.4 months. RNA samples were extracted using standard methods. The use of tissue samples and medical record data was approved for each patient cohort by the individual Institutional Review Boards. This project was approved by the University of New South Wales Human Research Ethics Committee (HC12551).

### 2.2. Cell Lines and Maintenance

The EOC cell lines HEY, PEO4 and JAM were kind gifts from Georgia Chenevix-Trench (QIMR Berghofer Medical Research Institute, Brisbane, QLD, Australia), while the 27/87 and SKOV3 cells were kindly provided by Anna deFazio (Westmead Millennium Institute for Medical Research, Westmead, NSW, Australia). The patient-derived WEHI-CS62 EOC cell line was first described by Topp et al., (2014) [24]. The non-malignant human ovarian surface epithelial (HOSE6.3) cells were kindly provided by Caroline Ford (Lowy Cancer Research Centre, Kensington, NSW, Australia).

The WEHI-CS62 cells were grown in DMEM F12 (Gibco, Waltham, MA, USA) with 10% foetal bovine serum (FBS), 1 µg/mL of hydrocortisone (Sigma, St. Louis, MI, USA), 8 µg/mL of insulin (Sigma, St. Louis, MI, USA) and 0.05 µg/mL of epidermal growth factor (Sapphire Bioscience, Redfern, NSW, Australia). The HOSE6.3 cells were cultured in media 199 and MCDB 105 media mixed in a 1:1 ratio with 10% FBS and 1× penicillin–streptomycin–glutamine. All other cell lines were maintained in RPMI media (Gibco, Waltham, MA, USA) with 10% FBS. Cells were split twice a week using trypsin, with the exception of WEHI-CS62 EOC cells for which Puck’s EDTA (140 mM NaCl, 5 mM KCl, 5.5 mM glucose, 4 mM NaHCO_3_, 0.8 mM EDTA and 9 mM HEPES) was used. All cell lines were STR profiled to be a 100% match to the reference samples and were mycoplasma free, tested every 6 months using a MycoAlert mycoplasma detection kit (Lonza, Basel, Switzerland).

### 2.3. Quantitative PCR (qPCR)

Reverse transcriptase (RT)-qPCR was performed to determine the gene expression levels of ABCA1 with the 7900HT Fast Real-Time PCR system (Applied Biosystems, Waltham, MA, USA). RNA was reverse transcribed using MMLV reverse-transcriptase (Life Technologies, Sydney, NSW, Australia) as previously described [23]. The mRNA expression levels of ABCA1 were determined using the TaqMan^®^ gene expression assay (ID: Hs00194045_m1) in a 96-well plate format (20 ng cDNA (mRNA equivalent) per well). Gene expression levels were quantified in relation to the expression of control genes (HPRT, ID: Hs99999909_m1, and GUSB, ID: Hs99999908_m1) using the ΔΔCt method, and the average of the expression values was calculated. Relative gene expression levels were normalized by log_2_ transformation.

### 2.4. Western Blots

Protein extraction and Western blotting were performed as described in Jung et al., (2020) [22]. Minor changes included incubation of the spheroids in radioimmunoprecipitation buffer for 30 min and 20–50 μg/well of protein were loaded onto SDS-PAGE gels.

The antibodies used were against ABCA1 (mouse monoclonal (AB.H10) #Ab18180; Abcam, Cambridge, UK; 1:750 to 1:1000), ß-actin (rabbit polyclonal #A2066; Sigma-Aldrich, St. Louis, MI, USA; 1:2000 to 1:3000, anti-rabbit-HRP #A0545; Sigma-Aldrich, St. Louis, MI, USA; 1:10,000) or anti-mouse-HRP (#NXA931; VWR, Radnor, PA, USA; 1:5000). All antibodies were diluted in in 5% skim milk except for anti-ß-actin antibody (diluted in 2.5% bovine serum albumin). Signals were detected using Clarity ECL Western Blotting Substrate (BIO-RAD, Hercules, CA, USA).

### 2.5. siRNA Transfections

HEY, 27/87 and JAM cells at 50–60% confluency were transfected with 10 nM of ABCA1-specific siRNAs (siRNA-1 5′ GGAGAUGGUUAUACAAUAGUUUU 3′; siRNA-2 GAAGAAAACUGGUGUCUAU, Dharmacon, Lafayette, CO, USA) or a non-targeting control siRNA (5′ GCACTACCAGAGCTAACTCAGATAGTACT 3′, Dharmacon) using Lipofectamine2000 (Gibco, Waltham, MA, USA) according to manufacturer’s instructions. WEHI-CS62 cells were transfected with the same conditions with the exception that Lipofectamine RNAimax (Gibco) was used as the delivery vehicle.

### 2.6. Growth Curves

HEY and 27/87 cells were plated at 24 h post-transfection, at 1 × 10^5^ cells/well in 6-well plates and harvested and counted at the indicated time points. 

### 2.7. Bromodeoxyuridine (BrdU) Incorporation Assays

BrdU incorporation assays were performed using the Bromodeoxyuridine (BrdU) Incorporation Assay Kit (Sigma-Aldrich, St. Louis, MI, USA) according to the manufacturer’s instructions, except for the inclusion of a post-fixation blocking step in PBS containing 10% FBS. 

### 2.8. Apoptosis Detection

Apoptosis was assessed using Annexin V and propidium iodide staining as described in Gao et al., (2020) [25]. Assays were performed for HEY and 27/87 cells at 72 h post-transfection.

### 2.9. Wound Healing Assay

Cell migration was measured using wound healing assays as described in Henderson et al., (2011) [26]. Minor changes included seeding of cells 24 h post-transfection at 4 × 10^4^ cells per chamber of Culture-Inserts (Ibidi), and the inserts were removed 16 h post-seeding for HEY and 24 h post-seeding for the 27/87 cell lines and photographed 4.5 h and 16 h later, respectively.

### 2.10. Growth in 3D Spheroids

Cell growth in 3D was assessed by measuring the volume and/or cell number in the spheroids. The EOC cells were seeded into 96-well ultra-low adhesion plates (ULA plates, Corning, Corning, NY, USA) at densities of 8 × 10^3^ cells/well for the HEY, JAM and WEHI-CS62 cell lines and 6 × 10^3^ cells/well for the 27/87, PEO4 and SKOV3 cell lines. When assessing the effects of the ABCA1 suppression, seeding was performed at 24 h post-transfection. 

To assess spheroid volume, images of the spheroids were taken using the Olympus BX53 microscope at 50× magnification at the indicated time points. The circular area was measured using ImageJ software before radii and volume were extrapolated using the formulae r = √(area/π) and V = 4/3 (πr^3^), respectively. Cell number per spheroid was determined by dissociating spheroids in trypsin for 2–5 min with vortexing before cells were counted with haemocytometers.

To assess the effect of chemical agents on spheroid growth, HEY and HOSE6.3 cells were seeded onto ULA plates as described above and incubated for 24 h before the indicated doses of salinomycin, haloperidol, eicosapentaenoic acid (EPA), dasatinib or DMSO were added. Spheroids were incubated for a further 72 h before being photographed, dissociated, and the number of cells per spheroid counted and spheroid volume determined. 

### 2.11. Gene Set Enrichment Analysis

Expression data were downloaded from The Cancer Genome Atlas (TCGA) using the publicly available high-grade serous data set (*n* = 498). Patient tumours were stratified into either high (upper quartile; *n* = 122) or low (lower quartile; *n* = 123) expression of ABCA1 in comparison to the entire cohort across 18,462 annotated genes. Normalized expression values were used as input for a Gene Set Enrichment Analysis (GSEA) using software developed by the Broad Institute. Analysis was performed using Molecular Signatures Database Hallmark (v7.1) and C5:GO (v7.1) to identify significantly enriched gene sets between “high” (top 25% of the overall cohort) and “low” (bottom 25% of the overall cohort) expressing tumour samples of ABCA1. A gene set was considered to be significant if the gene set had a false discovery rate (FDR) *q*-value < 0.1.

### 2.12. Cholesterol Extraction and Quantification

Cholesterol was extracted and quantified using the cholesterol/cholesteryl ester assay kit from Abcam according to manufacturer’s instructions. Minor modifications included harvesting at least 1 million cells per sample, incubation of the samples in a buffer of chloroform, isopropanol and NP-40 in a ratio of 7:11:0.1, respectively, for 1–2 h at room temperature together with shaking and addition of cholesterol esterase to the reaction mix to measure total cholesterol. Cholesterol quantification was performed for HEY, JAM and 27/87 cells at 48 h after transfection with methyl-B-cyclodextrin (MBCD) or vehicle (water) added 24 h post-transfection to assess the effects of ABCA1 suppression and the effectiveness of MBCD in removing excess intracellular cholesterol. To conduct experiments with WEHI-CS62 cells at relevant levels of ABCA1, the cells were cultured in 3D, since ABCA1 expression levels increase under these conditions. Cells were seeded into ultra-low adhesion plates at 24 h post-transfection with or without MBCD and incubated for a further 48 h prior to harvesting. The level of cholesterol was expressed as a fold change relative to control siRNA transfected cells.

### 2.13. Cell Viability Assay

The HEY and HOSE6.3 cells were seeded into 96-well plates at 1 × 10^3^ and 8 × 10^3^ cells/well, respectively, to achieve 40–50% confluency 24 h later, at which point escalating doses of haloperidol, salinomycin or vehicle (DMSO) were added. Cells were treated with the agents for 72 h before they were incubated with resazurin-based reagent (Sigma-Aldrich) at 10% of media volume for 3–5 h. The Benchmark Plus plate reader (BIO-RAD) was used at 570–595 nm to measure percent viability normalised to DMSO-treated cells. IC50s were extrapolated by fitting nonlinear regression curves using GraphPad Prism v8.

### 2.14. General Statistical Analysis

Comparisons of ABCA1 gene expression values between EOC tumour samples and cell lines were made using one-way ANOVA. At least three independent runs were performed for all experiments unless otherwise specified. *p*-Values for data expressed as a fold-change relative to control siRNA-treated or DMSO-treated cells were derived from one-sample *t*-tests, with the exception of experiments involving MBCD treatment. For experiments involving MBCD treatment, *p*-values were derived from two-way ANOVA with Tukey’s multiple comparison test. For other experiments with multiple comparisons, one-way ANOVA with Dunnett’s multiple comparison test was performed. All statistical tests for in vitro experiments were performed using GraphPad Prism v8. All results represent the mean ± standard deviation (SD). *p*-Values < 0.05 were considered significant.

## 3. Results

### 3.1. ABCA1 Suppression Impaired Malignant Phenotypes of Epithelial Ovarian Cancer Cells

Since high ABCA1 expression is associated with worse outcome in EOCs [23], we wanted to identify cell lines that expressed levels of ABCA1 representative of tumours from patients with poor outcomes. We therefore evaluated the levels of ABCA1 expression in a panel of EOC cell lines using RT-qPCR and compared them to those in a cohort of EOC patient tumours (150 serous and 80 endometrioid tumours). A subset of cell lines, including 27/87, PEO14, HEY, A2780 and OVCAR3, exhibited levels of ABCA1 expression above or close to the median expression level of the patient cohorts (Figure 1A). Regarded as being representative of the major EOC subtypes and being suitable for in vitro assays, the serous HEY and high-grade endometrioid 27/87 cell lines were chosen from among these lines for further experimentation (Figure 1A). To investigate the biological role of ABCA1 in these EOC cells, ABCA1 expression was suppressed with each of two independent ABCA1-specific siRNAs (Figure 1B and Appendix A. ABCA1 suppression reduced the clonogenicity and growth of both cell lines (Figure 1C,D). The BrdU incorporation assays revealed a significant reduction in the proliferation of HEY cells in response to the loss of ABCA1 (Figure 1E). A similar trend was observed for 27/87 cells, but the proliferation decrease induced by siRNA-2 did not reach statistical significance (Figure 1E). Based on annexin V staining, the 27/87 cells showed significant increases in the proportion of apoptotic cells at 48 h post-transfection with ABCA1-specific siRNAs (Figure 1F). ABCA1 suppression also induced apoptosis in HEY cells, but the effect did not reach statistical significance (Figure 1F). Thus, ABCA1 suppression negatively affected proliferation and enhanced apoptosis, but the extent of these effects differed between the two EOC cell types. 

The impact of ABCA1 suppression on EOC cell motility was also investigated by monitoring the ability of cells to move across an artificial “wound” following transfection of ABCA1 siRNA. The duration of migration assays for each cell line was limited to ensure minimal impact of cell doubling on wound closure rate (Appendix A). ABCA1 suppression reduced the wound closure ability of HEY cells by 20–35% (Figure 2A; *p* = 0.029; 0.009) and of 27/87 cells by 40–60% (Figure 2B; *p* = 0.001), indicating that ABCA1 expression contributes to EOC cell migration.

### 3.2. ABCA1 Suppression Impaired the Development of Three-Dimensional Structures

Cultured EOC cells were found to possess characteristics more representative of patient tumours when allowed to form three-dimensional spheroids compared to cells cultured as monolayers [27,28,29]. Therefore, the impact of ABCA1 suppression on spheroid characteristics was investigated. Depletion of ABCA1 in HEY or 27/87 cells led to significantly reduced spheroid size and cell numbers per spheroid by comparison with control cells (Figure 3A,B and Appendix A. Given this result with well-established cell lines, we extended the study to an early passage patient-derived high-grade serous EOC cell line that was refractory to chemotherapy, WEHI-CS62 [24]. Interestingly, ABCA1 protein expression was enhanced when WEHI-CS62 cells were maintained in suspension culture (Appendix A). ABCA1 suppression also led to reduced spheroid volume and cell number in this cell line (Figure 3C). Conversely, when ABCA1 expression was induced upon exposure of cells to liver X receptor agonist T0901317 [30], the formation of larger spheroids was observed in WEHI-CS62, HEY and 27/87 cells (Figure 3D–F). In HEY cells, suppression of ABCA1 by siRNA could be partially reversed by treatment with T0901317 (Figure 3E, upper-left panel) and this, in turn, led to restoration of spheroid size (Figure 3E, bottom panels). T0901317 did not restore ABCA1 expression in the 27/87 cells, and this was consistent with the lack of restoration in spheroid size (Figure 3F, top-left and bottom panels). Together, these data indicate that high ABCA1 expression supports EOC cell growth in three-dimensional structures.

### 3.3. ABCA1 Suppression Induced Cholesterol Accumulation in EOC Cells

To gain mechanistic insight into the significance of high ABCA1 expression in EOC cell biology, gene set enrichment analysis (GSEA) was conducted using a publicly available high-grade serous EOC data set from TCGA (*n* = 498) for which high ABCA1 expression was associated with poor outcome [21]. Genes that were differentially expressed between EOC tumours with high and low ABCA1 expression levels were determined and enriched to identify pathways and biological processes related to high ABCA1 expression in EOC. Genes involved in cholesterol homeostasis appeared among the most highly ranked cancer hallmark pathways (Figure 4A) and also in the top five biological processes associated with high ABCA1 expression (namely, regulation of mononuclear cell migration, osteoclast differentiation, B-cell activation involved in immune response, positive regulation of sterol transport and vascular endothelial growth factor receptor signalling pathways; Figure 4B and Appendix A). This analysis suggests that high expression levels of ABCA1 could be impacting upon the cholesterol pathway in poor outcome tumours.

To investigate whether ABCA1 functions as a cholesterol transporter in EOC cells, intracellular cholesterol levels were measured upon ABCA1 suppression. Cholesterol accumulation was observed in both HEY and WEHI-CS62 serous ovarian cancer cell lines but not in the high-grade endometrioid 27/87 cell line (Figure 5A). Hence, the impact of ABCA1 suppression-mediated cholesterol accumulation on serous HEY and WEHI-CS62 cells was further examined.

To test whether disturbance of ABCA1-mediated intracellular cholesterol balance was an underlying factor in impairing the growth of serous EOC spheroids, the effect of depleting intracellular cholesterol using methyl-B-cyclodextrin (MBCD) in the face of ABCA1 suppression was examined. Indeed, MBCD offset the accumulated cholesterol from EOC cells following ABCA1 suppression, reducing their cholesterol content to levels similar to control cells (Appendix A). In turn, cholesterol depletion by MBCD restored the size of HEY and WEHI-CS62 spheroids to levels observed in control-siRNA-treated cells (Figure 5B), resulting in an increase in the number of cells per spheroid (Figure 5C).

### 3.4. Blocking Cholesterol Efflux Using FDA-Approved Agents Impaired Malignant Phenotypes in EOC Cells

The ability of ABCA1-mediated cholesterol accumulation to impair the growth of serous EOC cells raises the question of whether pharmacological agents that mimic this effect may be efficacious against EOC. Salinomycin and haloperidol are amongst several therapeutic interventions reported to induce cholesterol accumulation and to dampen aggressive phenotypes in triple-negative breast cancers [20]. The IC50 concentrations of haloperidol and salinomycin in HEY cells were determined to be 48.72 µM and 11.13 µM, respectively (Figure 6A), and the selectivity for tumour cells was apparent from similar assays testing cytotoxicity to non-malignant HOSE6.3 cells in which the IC50 values were determined to be substantially higher (Appendix A). Exposure of HEY cells to either drug at its IC50 or a sublethal dose (i.e., 30 µM haloperidol; 5 µM salinomycin) induced a significant rise in intracellular cholesterol (Figure 6B), in line with that observed upon ABCA1 siRNA suppression (Figure 4A). In addition, exposure of HEY cells cultured as 3D spheroids to either salinomycin or haloperidol resulted in a dose-dependent reduction in cell numbers (Figure 6C). Similar anti-growth effects of these agents were observed in WEHI-CS62 cells (Figure 6D). To determine if these agents are less cytocidal on non-malignant cells, we performed resazurin-based viability assays upon non-malignant human ovarian surface epithelial (HOSE6.3) cells. The IC50 of haloperidol at 153.9 µM was significantly higher compared to that of HEY cells, and that of salinomycin was somewhat higher (IC50 = 17.37 µM, Appendix A).

It was recently discovered in triple-negative breast cancer models that a combination of dasatinib and eicosapentaenoic acid (EPA) can induce cholesterol accumulation, resulting in a greater loss of cancer cell viability compared to single agents alone [19]. Dasatinib has been tested clinically for platinum-resistant EOC patients with 15% of patients showing a complete response [29], suggesting that combining this drug with EPA may provide an added benefit. Treatment of HEY and WEHI-CS62 cells with either dasatinib or EPA significantly reduced growth in 3D culture for both cell lines, although no further potentiation was observed with the two agents in combination (Figure 6C,D). This suggests that the addition of EPA does not further enhance the effects of dasatinib in these models. Taken together these data support the potential of salinomycin and haloperidol as therapeutic options for patients with serous EOC and high ABCA1 expression.

## 4. Discussion

Five-year survival rates of EOC patients have not improved over several decades indicating the urgent need for new highly selective, molecular-targeted therapies for this disease. The adipose-rich greater omentum is a preferred site of dissemination where EOC cells display increased cholesterol uptake and altered cholesterol and lipid metabolism. Many studies have shown that elevated levels of cholesterol or lipids can enhance cancer aggressiveness. However, this study demonstrates the critical importance of the ABCA1 transporter in removing intracellular cholesterol and maintaining cholesterol homeostasis to enable rapid cell growth and motility. This work also highlights several FDA-approved agents currently used for other diseases that may prove effective against EOC cells due to the fact of their abilities to induce cholesterol accumulation. Thus, ABCA1 is a potential therapeutic target for EOC and repurposing drugs known to induce intracellular cholesterol accumulation may be useful in the treatment of this aggressive disease.

This research helps elucidate why high ABCA1 expression has been previously linked to poor clinical outcome in this disease [23]. There are conflicting pieces of evidence demonstrating the biological roles of ABCA1 in cancer biology. While certain studies have shown that high expression of ABCA1 promotes migration and growth of human prostate and breast cancer cells [21,22,31], others have shown that ABCA1 suppresses the growth of p53 wild-type colorectal tumours and oral cancer cells [30,32] or that ABCA1 hypermethylation could be associated with increased growth of EOC cells for some cases [33]. The present study characterizes in detail how high-level ABCA1 expression contributes to EOC malignancy through enhancing cell growth and motility. This is the first report describing the need to maintain cholesterol balance as a potential Achilles heel in EOC. Human serous EOC cells have a preference to metastasize to the omentum due to the homing adipokines, such as IL6 and IL8, produced by the omental adipocytes, and they are thought to uptake lipids from the adipocytes and shift towards oxidative phosphorylation for energy production [8]. EOCs are also known to possess elevated levels of cholesterol compared to non-malignant ovarian epithelial cells [12]. Such observations have fuelled the concept that in EOC biology, cholesterol acts mainly to promote tumour progression. However, excessive levels of cholesterol in macrophages have long been known to induce cytotoxicity [20,34,35,36]. The new data presented here point to the existence of an upper threshold for intracellular cholesterol in serous EOC cells, beyond which cholesterol can limit growth or migration. Similar observations have been made regarding ABCA1 suppression in triple-negative breast cancers [21,22], and together with the current study, indicate that for cancers that thrive in cholesterol-rich environments, disrupting cholesterol efflux may be a viable therapeutic strategy.

One possible mechanism by which cholesterol accumulation can affect serous EOC growth relates to the ratio between cholesterol and phospholipids, which determine the fluidity of cell membranes and the ability for cells to change shape and migrate. In triple-negative breast cancer, elevated levels of cholesterol in cell membranes can reduce membrane fluidity, leading to problems with cell migration [22]. Elevated intracellular cholesterol has also been reported to induce apoptosis through various pathways. For example, Fas/CD95 death receptors can aggregate in lipid rafts of cholesterol-rich cells to trigger apoptosis, a phenomenon that is reversible through MBCD-mediated cholesterol depletion [37,38]. Stress from cholesterol overload in the endoplasmic reticulum or mitochondria can also trigger apoptosis [35,38,39,40]. Future investigation into whether the cholesterol accumulation observed in EOC cells impairs cell growth and migration through any of these avenues could, in turn, provide additional approaches for targeted therapy. 

As ABCA1 is not the only transporter with cholesterol efflux function, more potent effects on cholesterol accumulation may be achieved by targeting other cholesterol transporters concurrently. ABCG1, for example, is known to efflux cholesterol, and expression of both ABCA1 and ABCG1 are needed to deplete lipid rafts in tumour-associated macrophages [41]. Identifying the transporters that can compensate for the loss of ABCA1 in EOC will determine which transporters might best be targeted alongside ABCA1 to potentiate the anti-oncogenic effects of ABCA1 suppression.

By discovering the ability of cholesterol accumulation to impair the growth of serous EOC cells, this study raises the question of whether agents that exert such functions can be an effective treatment for these EOCs. Indeed, here we show that haloperidol and salinomycin can slow the growth of serous EOC cells. Since these compounds are approved by the Food and Drug Administration (FDA), they are potentially adaptable for clinical translation in EOC patients. While no studies have reported the use of haloperidol or the combination of dasatinib and EPA on EOC cells, salinomycin and dasatinib as single agents have been shown to reduce the aggressiveness of EOCs. Whilst this study also found that dasatinib can exert potent anti-oncogenic functions in serous EOC cells, no further potentiation of its efficacy by EPA was observed in the cell lines studied here. This was in contrast to observations made in triple-negative breast cancer, where the combination of EPA with dasatinib led to an accumulation of intracellular cholesterol as a potential explanation for the heightened anti-growth effects observed in triple-negative breast cancer models [21]. The absence of evidence for this drug combination to induce cholesterol accumulation in EOC cells may be accountable for the lack of potentiation. Overall, we have shown that despite differing modes of action and indications of the four agents tested in this study, the ability of salinomycin and haloperidol to induce cholesterol accumulation enables them to elicit selective, anti-growth effects on EOC cells.

While our study primarily focused on the cholesterol efflux functions of ABCA1, it is important to note that the data in the 27/87 cell line suggests ABCA1 may contribute to EOC biology independent of its cholesterol efflux functions. Since no accumulation of cholesterol was observed in this cell line, despite growth and migration being stunted by ABCA1 suppression, it is possible that reduced efflux of ABCA1 substrates other than cholesterol, such as sphingosine-1-phosphate, may be responsible for these effects in this case [41,42,43,44]. Conversely, since our data showed that the contributions of ABCA1 to the malignancy of 27/87 is cholesterol independent and our focus was on this particular function of ABCA1, we have not investigated the impact of inducing cholesterol accumulation in this cell line. If inducing cholesterol accumulation in this cell line is cytocidal, compounds such as haloperidol or salinomycin could be investigated as potential treatments for this endometrioid EOCs.

## 5. Conclusions

In conclusion, suppression of ABCA1 can reduce the growth and migration of EOC cells, and in serous EOC cells these effects are, in part, caused by intracellular cholesterol accumulation. Altogether, the data presented here demonstrate that ABCA1 can support the growth of serous EOC cells by maintaining cholesterol homeostasis, which may be an underlying explanation as to why serous EOC patients with elevated tumoral expression of ABCA1 were found to have inferior survival rates. ABCA1 is thus a potential therapeutic target for this disease and FDA-approved agents that induce cholesterol accumulation should be further investigated for EOC.

## Figures and Tables

**Figure 1 cancers-14-01878-f001:**
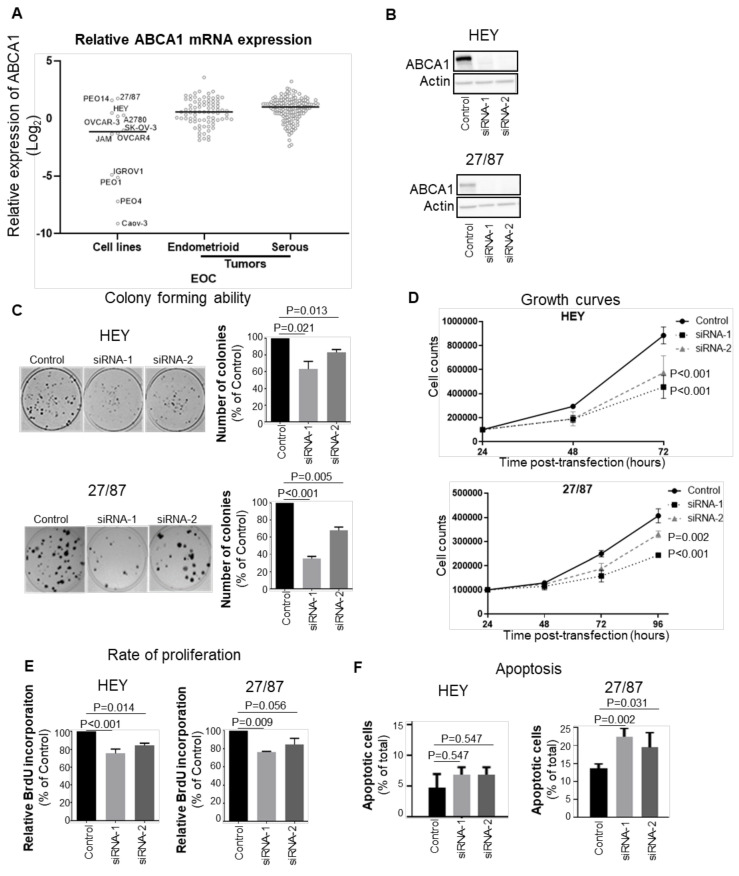
ABCA1 is required for rapid growth of epithelial ovarian cancer (EOC) cells in monolayer. (**A**) RT-qPCR assessment of ABCA1 expression across a panel of cell lines and ovarian cancer tumours. ABCA1 mRNA expression was normalized to the control genes, GUSB and HPRT, and graphed relative to ABCA1 expression in PEO4. Black represents the median ABCA1 expression of the group. (**B**) Western blots showing the extent of the knockdown of ABCA1 by two independent siRNAs in the serous HEY and endometrioid 27/87 cell lines. (**C**) Suppression of ABCA1 impaired the colony forming ability of EOC cells. *n* = 3. *p*-Values were derived from one-sample *t*-tests. (**D**) ABCA1 knockdown reduced the growth of EOC cells, determined by cell counting. *n* ≥ 3. *p*-Values were derived from to-way ANOVA with Dunnett’s multiple comparison tests. (**E**) ABCA1 knockdown by siRNA-1 and siRNA-2 reduced the rate of EOC cell proliferation. *n* = 4 for HEY, *n* = 3 for 27/87. *p*-Values were derived from one-sample *t*-tests. (**F**) ABCA1 knockdown by siRNA-1 and siRNA-2 increased apoptosis in the 27/87 cells as measured by Annexin V and propidium iodide staining. *n* = 3. *p*-Values were derived from two-way ANOVA with Dunnett’s multiple comparison test. All results represent the mean ± standard deviation (SD).

**Figure 2 cancers-14-01878-f002:**
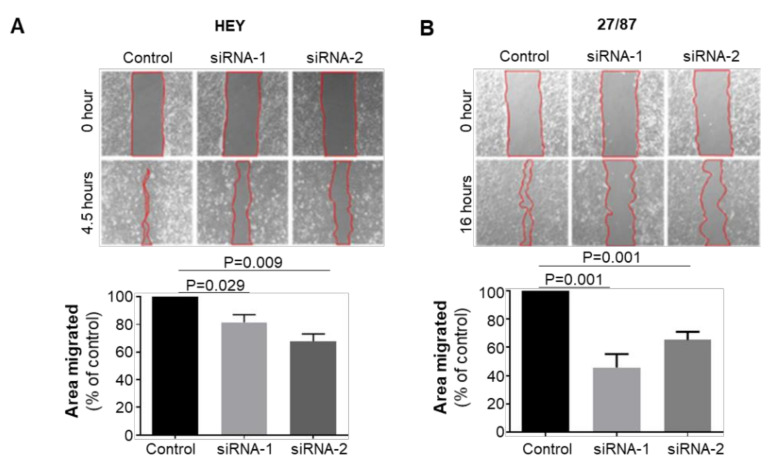
ABCA1 supported the migration of epithelial ovarian cancer cells. Wound healing assays performed on (**A**) HEY and (**B**) 27/87 EOC cells with or without ABCA1 suppression. Changes in relative wound size was expressed as a percentage of control. *n* = 3 (**A**), *n* = 4 (**B**). Results represent the mean ± SD. *p*-Values were derived from one-sample *t*-tests.

**Figure 3 cancers-14-01878-f003:**
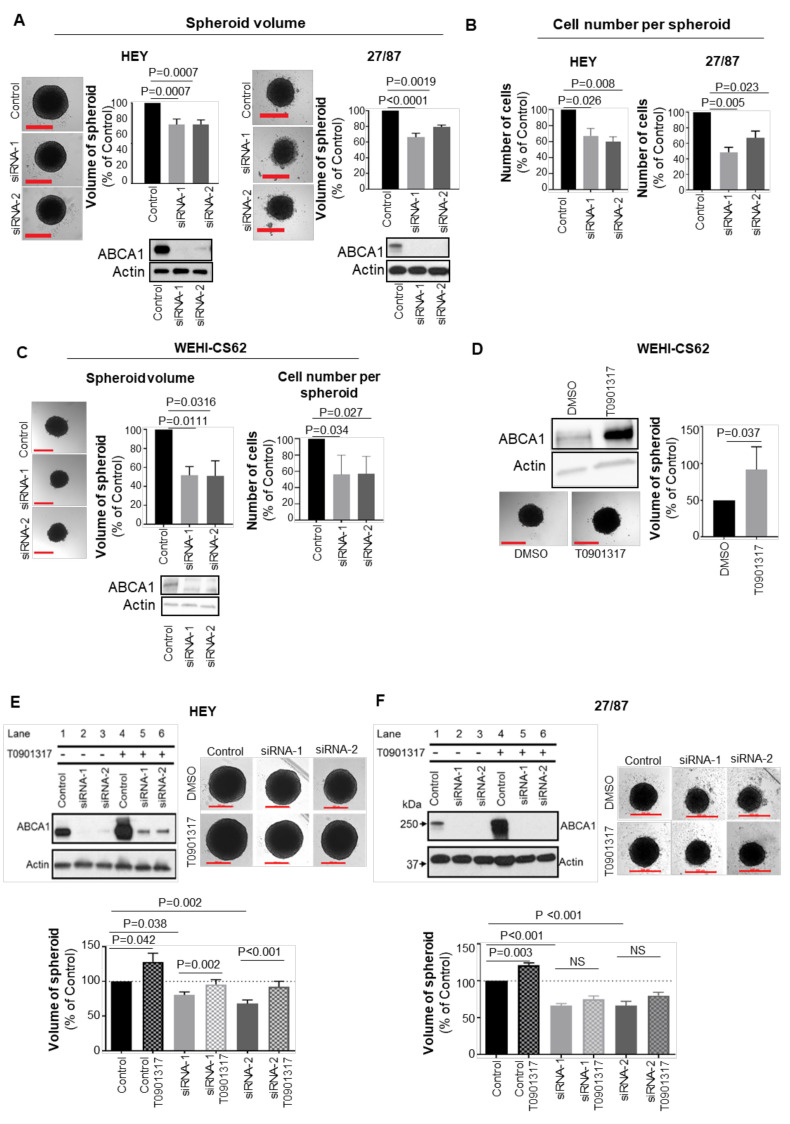
ABCA1 suppression reduced the size of epithelial ovarian cancer (EOC) spheroids. (**A**) Representative photographs (left panels) and column graphs (right panels) showing the effect of ABCA1 suppression on the size of HEY and 27/87 EOC spheroids at 72 h after seeding into low-adhesion plates and 96 h after transfection. *n* = 3. *p*-Values were derived from one-sample *t*-tests. (**B**) Changes in the number of cells per spheroid after ABCA1 suppression in HEY and 27/87 cells. Assays performed at the same time as for (**A**). *n* = 3. *p*-Values were derived from one-sample *t*-tests. (**C**) Representative photographs (left panels) and column graphs (right panels) showing the effect of ABCA1 suppression on the size of WEHI-CS62 patient-derived, high-grade serous EOC spheroids at 72 h after seeding into low-adhesion plates and 96 hours after transfection. *n* = 3. *p*-Values were derived from one-sample *t*-tests. (**D**) The effect of ABCA1 overexpression, achieved using T0901317, on WEHI-CS62 spheroid size. *n* = 3. *p*-Values were derived from unpaired, two-tailed *t*-tests. (**E**) The effect of ABCA1 re-expression, achieved using T0901317, on HEY spheroid size. *n* = 3. (**F**) The effect of ABCA1 re-expression, achieved using T0901317, on 27/87 spheroid size. *n* = 3. All results represent the mean ± SD. *p*-Values derived from one-way ANOVA with Dunnett’s multiple comparison tests. Scale bars represent 500 µm.

**Figure 4 cancers-14-01878-f004:**
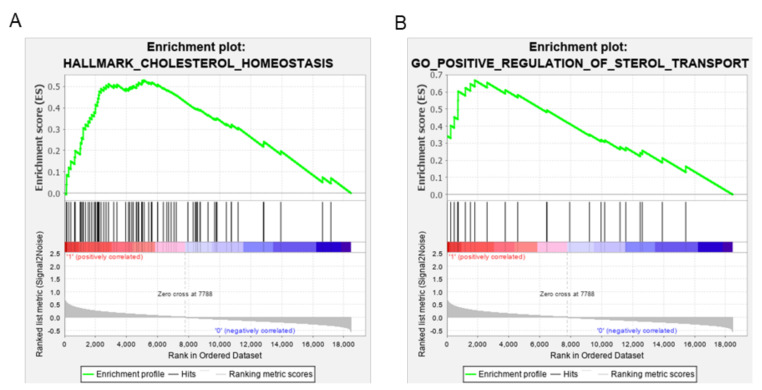
Gene set enrichment analysis (GSEA) showing EOC tumours with high ABCA1 are enriched for expression of gene sets associated with cholesterol metabolism. (**A**) GSEA showing enrichment in the Hallmark gene set involved in cholesterol homeostasis. *p* = 0.008 and false discovery rate (FDR) = 0.021. (**B**) GSEA showing enrichment in the Gene Ontology (GO) gene set involved in positive regulation of sterol transport. *p* < 0.001 and FDR = 0.069. *n* = 498.

**Figure 5 cancers-14-01878-f005:**
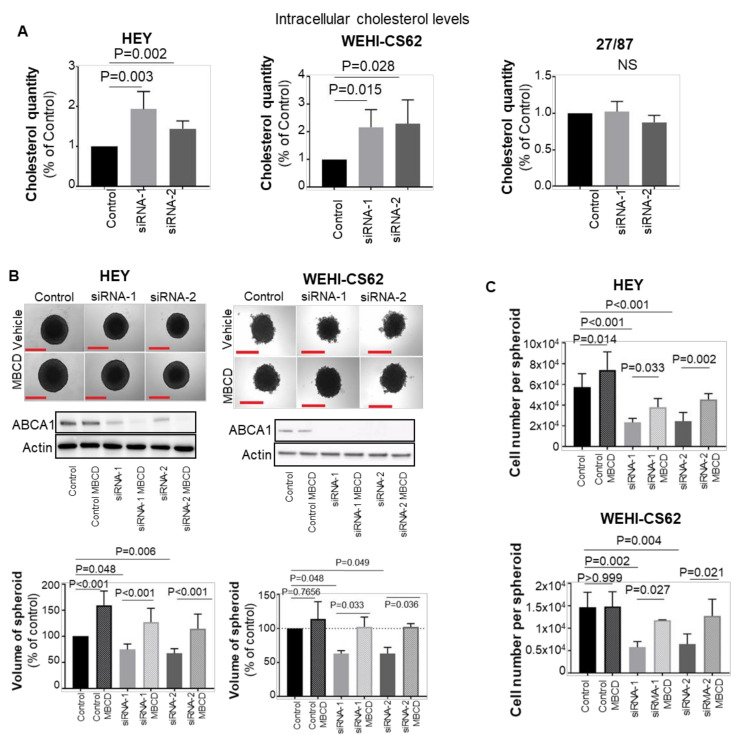
ABCA1 suppression induced cholesterol accumulation in serous ovarian cancer cells which was associated with decreased growth in the EOC cells. (**A**) Changes in intracellular cholesterol levels following ABCA1 suppression. *n* = 5 for serous HEY and WEHI-CS62 cells. *n* = 3 for endometrioid 27/87 cells. (**B**) Cholesterol depletion using methyl-B-cyclodextrin (MBCD) reversed the effect of ABCA1 suppression on 3D growth characteristics of HEY (upper panels) and WEHI-CS62 (lower panels). Experiments were performed at 144 h in 3D culture or 168 h after transfection for HEY and 72 h in 3D culture or 96 h after transfection for WEHI-CS62. *n* = 5 for HEY and *n* = 3 for WEHI-CS62 cells. (**C**) Changes in the number of cells per spheroid after ABCA1 suppression with or without MBCD treatment performed at same time points as for (**B**). *n* = 5 for HEY and *n* = 3 for WEHI-CS62 cells. All results represent mean ± SD. *p*-Values for (**A**) were derived from one-sample *t*-test, whilst all other *p*-values were derived from two-way ANOVA with Tukey’s multiple comparison tests. Scale bars represent 500 µm.

**Figure 6 cancers-14-01878-f006:**
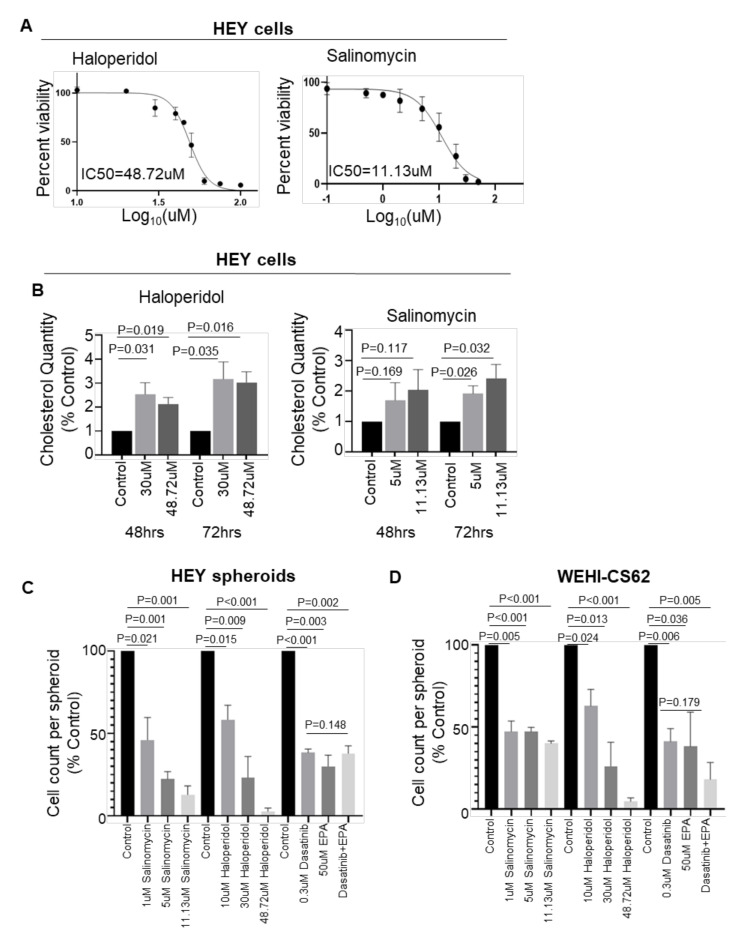
The effect of agents reported to induce cholesterol accumulation on EOC spheroid formation and cell growth. (**A**) Dose–response curves generated by resazurin-based assays showing the half maximal inhibitory concentrations (IC50) of haloperidol and salinomycin upon malignant serous EOC HEY cells after 72 h treatment. Cells were cultured in monolayer conditions. Percent viability is the viability of the drug-treated cells expressed as a percentage of the viability of DMSO-treated cells. (**B**) Cholesterol quantification after 48–72 h showing the effect of salinomycin and haloperidol in HEY cells. Cells were cultured in monolayer conditions. (**C**) Column graphs represent changes in the cell number per spheroid formed by HEY cells after 72 h treatment with vehicle or one of the indicated drugs. (**D**) Column graphs represent changes in the cell number per spheroid formed by the WEHI-CS62 cells after treatment with vehicle or one of the indicated drugs. *n* = 3 for all experiments. *p*-Values for (**B**–**D**) were derived from one-sample *t*-tests with the exception of cell count comparisons between the dasatinib and EPA combination with single agents alone for which one-way ANOVA with Dunnett’s multiple comparison tests were performed. Results represent the mean ± SD.

## Data Availability

The data that support the findings of this study are available from the corresponding author (mhenderson@ccia.org.au) upon reasonable request.

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
