# Peer review of "Suppression of the ABCA1 Cholesterol Transporter Impairs the Growth and Migration of Epithelial Ovarian Cancer"

_cancers, 2022, doi:10.3390/cancers14081878_

Round 1
Reviewer 1 Report
In this manuscript authors have examined the role of ABCA1 cholesterol transported in epithelial ovarian cancer progression. The data presented by the authors supports the importance of increased level of ABCA1 in EOC. Authors performed various in vitro assays to support their arguments, which provides a strong correlation between EOC progression and ABCA1.
Specific Comments:
- Fig. 1F lines 257-258: the values are not significant in HEY. Authors should be specific when presenting the results and not vague. Please state clearly in which cell lines the results were significant and in which they were not.
- It wasn't clear that how did the authors say the level of ABCA1 are higher in EOC, compared to what? and is there any data which authors could provide or point to? what are the levels of ABCA1 in other cancer types compared to EOC? What are the base levels of ABCA1 to which the authors have compared to say that it is high in EOC?
- Can the authors perform IHC on a few human tissue samples representing EOC and others to make an argument that ABCA1 is specifically increased in EOC? currently, the data provided in figure 1 is transcript data and doesn't strongly supports the high expression argument made by the authors. Authors should provide more evidence to support their argument. Even the cell line transcript data is not in concordance with the WB (protein levels) show in the supplementary figure. eg. HEY has higher transcript but lower protein compared to PE04. therefore, it warrants some IHC samples to be able to strongly support the arguments.
- Fig 2A: Why did the authors didn't consider a Boyden Chamber Transwell migration and invasion assay? considering, that authors present the importance of ABCA1 in 3D spheroids. It would be great if authors could perform trans well migration assay.
- What is percent viability? how the values were normalized, please ellaborate in the legend for 6A.
Author Response
Re: Comments and Suggestions for Authors:
We have addressed all comments. The comments are highlighted in bold. Please see the attached PDF file for figures as figures cannot be inserted into this text box.
Reviewer 1:
Comment 1: Fig. 1F lines 257-258 (line 276-285 in current version): the values are not significant in HEY. Authors should be specific when presenting the results and not vague. Please state clearly in which cell lines the results were significant and in which they were not.
To address the comments of reviewer 1, we have added text to fully describe which phenotype was significant in each cell line as highlighted below:
Lines 276-285 (in current version):
‘BrdU incorporation assays revealed a significant reduction in proliferation in HEY cells in response to loss of ABCA1 (Fig.1E). A similar trend was observed for the 27/87 cells, but the proliferation decrease induced by siRNA-2 did not reach statistical significance (Fig.1E). Based on annexin V staining, 27/87 cells showed significant increases in the proportion of apoptotic cells at 48 hours post transfection with ABCA1-specific siRNAs (Fig. 1F). ABCA1 suppression also induced apoptosis in HEY cells but the effect did not reach statistical significance (Fig. 1F). Thus, ABCA1 suppression negatively affects proliferation and enhances apoptosis but the extent of these effects differs between the two EOC cell types.’
Comment 2: It wasn't clear that how did the authors say the level of ABCA1 are higher in EOC, compared to what? and is there any data which authors could provide or point to? what are the levels of ABCA1 in other cancer types compared to EOC? What are the base levels of ABCA1 to which the authors have compared to say that it is high in EOC?
In the manuscript there is no claim made that ABCA1 levels are higher in EOC than other cancers. Any mention of relative levels is within the patient cohort and within the cell line cohort and we apologise for any lack of clarity. We have previously published in Hedditch et al., 2014, JNCI doi: 10.1093/jnci/dju149 that ABCA1 expression is higher in EOC tumors than in non-malignant fallopian tube and that high-level ABCA1 expression in EOC tumors is associated with poor outcome. This, along with the need to maintain intracellular cholesterol homeostasis with efflux pumps, forms the rationale for examining ABCA1 in EOCs, not the fact ABCA1 is higher in this cancer compared to other malignancies.
We make the point that the cell lines were selected based on their ABCA1 levels being consistent with the levels observed in patients with poor outcome, that is, above the median of the patient cohort.
To address the query, we have clarified the statement made regarding our rationale for cell line selection at lines 263-273 as follows: (new text highlighted)
“Since high ABCA1 expression is associated with worse outcome in EOC [23], we wanted to identify cell lines that express levels of ABCA1 representative of tumors from patients with poor outcome. We therefore evaluated the levels of ABCA1 expression in a panel of EOC cell lines using RT-qPCR and compared them to those in a cohort of EOC patient tumours (150 serous and 80 endometrioid tumours). A subset of cell lines, including 27/87, PEO14, HEY, A2780 and OVCAR3, exhibited levels of ABCA1 expression above or close to the median expression level of the patient cohorts (Fig. 1A). Regarded as representative of the major EOC subtypes and being suitable for in vitro assays, the serous HEY and high grade endometrioid 27/87 cell lines were chosen from among these lines for further experimentation (Fig. 1A).”
In response to the Reviewer’s query about how ABCA1 expression compares across different tumour types, the analysis below shows measurable expression of ABCA1 RNA across cohorts of several epithelial tumour types (R2 database: https://hgserver1.amc.nl/cgi-bin/r2/main.cgi). This information has not been added to the manuscript.
Comment 3: Can the authors perform IHC on a few human tissue samples representing EOC and others to make an argument that ABCA1 is specifically increased in EOC? currently, the data provided in figure 1 is transcript data and doesn't strongly supports the high expression argument made by the authors. Authors should provide more evidence to support their argument. Even the cell line transcript data is not in concordance with the WB (protein levels) show in the supplementary figure. eg. HEY has higher transcript but lower protein compared to PE04. therefore, it warrants some IHC samples to be able to strongly support the arguments.
In a previous publication (Hedditch et al., 2014, JNCI doi: 10.1093/jnci/dju149), there is IHC data showing that ovarian cancer tumors express higher ABCA1 protein compared to non-malignant fallopian tube. As this is already published, there was no need to replicate this data.
As explained above in response to Comment 2, no claim is made of ABCA1 expression being higher in EOC than in other cancers. The point made, and supported by Hedditch et al., 2014, JNCI doi: 10.1093/jnci/dju149, is that the subset of patients with high ABCA1 expression (above the median expression for the cohort) do poorly compared to other EOC patients. We expect that the improved wording used in the paragraph above (lines 263-273) now addresses this issue.
The reviewer’s comment about expression of ABCA1 in HEY cells relative to PEO4 cells refers to data presented in Supplementary Figure S2. In this figure, the relative expression of ABCA1 between different cell lines cannot be compared because they are not from the same blot. The purpose of that figure is to indicate changes that can occur in ABCA1 levels when cells are cultured in 2D versus 3D growth conditions. This is why in the densitometry, the 2D monolayer is given an arbitrary value of 1 so that the fold change in ABCA1 levels in the 3D spheroids could be calculated. We have made no comment about the levels of ABCA1 protein across different cell lines with this figure. Since ABCA1 levels in EOC detected via IHC has already been published in our 2014 study, we believe it is not necessary here.
Comment 4: Fig 2A: Why did the authors didn't consider a Boyden Chamber Transwell migration and invasion assay? considering, that authors present the importance of ABCA1 in 3D spheroids. It would be great if authors could perform trans well migration assay.
The cell lines we have used do not migrate in the Transwell chamber efficiently. They form uncountable clumps and do not adhere to the Transwell membranes and hence counts cannot be accurate.
Comment 5: What is percent viability? how the values were normalized, please ellaborate in the legend for 6A.
Percent viability is the viability normalised to DMSO control. The values are calculated using this equation: (viability assay readout for the drug treatment ÷ readout for the DMSO control) x 100. These changes have been made to legend of figure 6A as follows:
(Lines 454-455):
Figure 6. The effect of agents reported to induce cholesterol accumulation on EOC spheroid formation and cell growth. (A) Dose-response curves generated by resazurin-based assays showing the half maximal inhibitory concentrations (IC50) of haloperidol and salinomycin upon malignant serous EOC HEY cells after 72 h treatment. Cells were cultured in monolayer conditions. Percent viability is the viability of the drug-treated cells expressed as percentage of the viability of DMSO-treated cells. (B) Cholesterol quantification after 48-72 hours showing the effect of salinomycin and haloperidol in HEY cells. Cells were cultured in monolayer conditions. (C) Column graphs represent changes in cell number per spheroid formed by HEY cells after 72 h treatment with vehicle or one of the indicated drugs. (D) Column graphs represent changes in cell number per spheroid formed by the WEHI-CS62 cells after treatment with vehicle or one of the indicated drugs. N=3 for all experiments. P-values for (B), (C) and (D) were derived from One-sample t-test with the exception of cell count comparisons between the dasatinib and EPA combination with single agents alone for which One-ANOVA with Dunnett’s multiple comparisons test was done. Results represent mean ± SD.

Reviewer 2 Report
The aim of this manuscript is to investigate the role of ABCA1 in cholesterol balance in Epithelial Ovarian Cancer (EOC). In this context, the authors evaluate the effects of ABCA1 suppression on different pathways: intracellular cholesterol balance, EOC cells’ growth and migration, also on the features of 3-dimensional EOC spheroids.
Even if the manuscript provides an organic overview, with a densely organized structure and based on well-synthetized evidence, there are aspects to be mentioned, to make the article fully readable. For these reasons, the manuscript requires minor changes.
Please find below an enumerated list of comments on my review of the manuscript:
INTRODUCTION:
LINE 66: There are some minor comments for this section. Ovarian cancer is a disease conventionally classified according to histologic subtype, each of them characterized by distinct clinicophatological and molecular features. Among these subtypes, EOC is included in an heterogeneous group of neoplasia: for these reasons, the paper will benefit from providing an organic description of these ovarian cancer subtypes, specifically in terms of clinical and morphological features of this disease (see, for reference: Giusti, I., Bianchi, S., Nottola, S. A., Macchiarelli, G., Dolo, V. (2019). CLINICAL ELECTRON MICROSCOPY IN THE STUDY OF HUMAN OVARIAN TISSUES. EuroMediterranean Biomedical Journal, 14).
LINE 88: Also recent studies have detected the expression levels of ABCA1, in human triple-negative breast cancer, significantly associated with the histological grade (see, for reference: Pan, H., Zheng, Y., Pan, Q., Chen, H., Chen, F., Wu, J., & Di, D. 2019. Expression of LXR‑β, ABCA1 and ABCG1 in human triple‑negative breast cancer tissues. Oncology Reports, 42(5), 1869-1877).
In conclusion, this manuscript is densely presented and well organized, based on well-synthetized evidences. The authors were lucid in their style of writing, making it easy to read and understand the message, portrayed in the manuscript. Besides, the methodology design was rigorous and appropriately implemented within the study. However, many of the topics are very concisely covered. This manuscript provided a comprehensive review of current knowledge in this field. Moreover, this research have futuristic importance and could be potential for future research. However, the minor concern of this manuscript is with the introductive section: for these reasons, I have minor comments only for the introductive section, for improvement before acceptance for publication. The article is accurate and provides relevant information on the topic and I suggest minor changes to be made in order to maximize its scientific impact. I would accept this manuscript, if the comments are addressed properly.
Author Response
Re: Comments and Suggestions for Authors:
We have addressed all comments. The comments are highlighted in bold.
Reviewer 2:
Comments:
LINE 66: There are some minor comments for this section. Ovarian cancer is a disease conventionally classified according to histologic subtype, each of them characterized by distinct clinicophatological and molecular features. Among these subtypes, EOC is included in an heterogeneous group of neoplasia: for these reasons, the paper will benefit from providing an organic description of these ovarian cancer subtypes, specifically in terms of clinical and morphological features of this disease (see, for reference: Giusti, I., Bianchi, S., Nottola, S. A., Macchiarelli, G., Dolo, V. (2019). CLINICAL ELECTRON MICROSCOPY IN THE STUDY OF HUMAN OVARIAN TISSUES. EuroMediterranean Biomedical Journal, 14).
LINE 88: Also recent studies have detected the expression levels of ABCA1, in human triple-negative breast cancer, significantly associated with the histological grade (see, for reference: Pan, H., Zheng, Y., Pan, Q., Chen, H., Chen, F., Wu, J., & Di, D. 2019. Expression of LXR‑β, ABCA1 and ABCG1 in human triple‑negative breast cancer tissues. Oncology Reports, 42(5), 1869-1877).
….However, the minor concern of this manuscript is with the introductive section: for these reasons, I have minor comments only for the introductive section, for improvement before acceptance for publication. The article is accurate and provides relevant information on the topic and I suggest minor changes to be made in order to maximize its scientific impact. I would accept this manuscript, if the comments are addressed properly.
We have now addressed these suggestions for improving the Introduction as follows:
Lines 68-75, we added a description of the features of the EOC subtypes:
“HG-SOC are thought to arise from the ciliated cells of the Fallopian tube, are highly proliferative and 80% of cases are TP53 mutant [3,4]. Low-grade serous EOCs are much rarer and usually do not exhibit TP53 mutations [3,4]. Endometrioid EOCs account for 10-20% of all EOCs and exhibit a range of somatic mutations including ARID1A and PTEN [3,4]. ARID1A mutations are also often seen in clear cell carcinoma which accounts for ~5% of all EOCs [3,4]. Mucinous is the rarest subtype, accounting for 2-3% of EOCs and characterized by KRAS mutations [3,4]. High-grade serous and endometrioid ovarian cancers have the poorest clinical prognosis compared to other subtypes [3,4]. “
Lines 87-88 and 99-103, we added references to the study showing high expression of ABCA1 in triple-negative breast cancer:
“Here, ovarian cancer cells can uptake cholesterol and other lipids from their extracellular environment resulting in altered lipid metabolism [9, 12, 13, 14].”
“In triple-negative breast cancer, another gynecological malignancy that thrives in a cholesterol-rich environment, impairing cholesterol efflux through suppression of the ATP-binding cassette A1 (ABCA1) cholesterol transporter leads to disruption of cholesterol homeostasis and impaired malignant characteristics [21, 22] and high expression of ABCA1 is associated with high grade tumours in this disease [14].”
The additional references have been added to the manuscript (lines 588-589 and 606-607).
- Giusti, I., S., Bianchi, S. A., Nottola, G., Macchiarelli, V. Dolo. CLINICAL ELECTRON MICROSCOPY IN THE STUDY OF HUMAN OVARIAN TISSUES. EMBJ 2019. 14(34): p. 145–151
- Pan, H., Y., Zheng, Q., Pan, H., Chen, F., Chen, J., Wu, & D. Di. Expression of LXR β, ABCA1 and ABCG1 in human triple negative breast cancer tissues. 2019 Oncology Rep., 42(5): p. 1869-1877.

Reviewer 3 Report
The authors show the implication of ABCA1 in the control of tumor growth and migration.
It would be desirable to have data points in all graphs.
Figure 1 C, images have different brightness. It would be nicer if the brightness is the same
In the text of figure 4 the authors talk about 5 top Biological processes but they show only GO term of positive regulation of sterol transport. It should be corrected.
Author Response
Re: Comments and Suggestions for Authors:
We have addressed all comments. The comments are highlighted in bold. All figures are in the attachment.
Reviewer 3:
Comment 1: The authors show the implication of ABCA1 in the control of tumor growth and migration. It would be desirable to have data points in all graphs.
In the manuscript figures, individual data points are shown in those graphs where this helps to visualise the data spread most effectively (eg Figure 1A). All other graphs that do not show individual data points instead include error bars, which indicate the standard deviation and thus the spread of the actual data across 3-4 independent replicate experiments. These error bars represent all the data within the experiment.
Therefore, we believe that showing individual data points in these graphs is not necessary and that the resulting delay in manuscript re-submission would not be justified. However, if the Editors disagree and the reviewer can specify which particular graphs would benefit then we could amend accordingly.
Comment 2: Figure 1 C, images have different brightness. It would be nicer if the brightness is the same.
This has been addressed and new images (please see attachment) have been inserted.
Figure 1
Reviewer 3 continued
Comment 3: In the text of figure 4 the authors talk about 5 top Biological processes but they show only GO term of positive regulation of sterol transport. It should be corrected.
The top 5 biological processes have been added as a Supplementary figure below (please see attachment):
Additional text has also been added at lines 370 and 376:
‘Genes involved in cholesterol homeostasis appeared among the most highly ranked cancer hallmark pathways (Fig. 4A) and also were among the top 5 Biological Processes associated with high ABCA1 expression (namely, Regulation of mononuclear cell migration, Osteoclast differentiation, B cell activation involved in immune response, Positive regulation of sterol transport, and vascular endothelial growth factor receptor signaling pathways; Fig. 4B and Fig. S3). This analysis suggests that high expression levels of ABCA1 could be impacting upon the cholesterol pathway in poor outcome tumours.’

Round 2
Reviewer 1 Report
Authors responded to the comments satisfactorily and have revised the manuscript accordingly. no more comments.